# Kinetics of Oil Absorption and Moisture Loss during Deep-Frying of Pork Skin with Different Thickness

**DOI:** 10.3390/foods10123029

**Published:** 2021-12-06

**Authors:** Hong-Ting Victor Lin, Po-Han Hou, Wen-Chieh Sung

**Affiliations:** 1Department of Food Science, National Taiwan Ocean University, Keelung 202301, Taiwan; HL358@mail.ntou.edu.tw (H.-T.V.L.); 10832006@mail.ntou.edu.tw (P.-H.H.); 2Center of Excellence for the Oceans, National Taiwan Ocean University, Keelung 202301, Taiwan

**Keywords:** pork skin, deep-frying, kinetic model, thickness

## Abstract

We have investigated different properties (thickness, moisture loss, oil uptake, breaking force, color, puffing ratio during 0.5–5 min frying, microstructure, and sensory evaluation) of raw pork skins with varying thickness (2, 3, and 4 mm) after drying, intended as deep-fried snacks. We have found that the oil content, breaking force, and puffing ratio of fried pork skin with different raw skin thickness have no significant difference under similar water content (1.68–1.98 g/100 g wet weight basis, wb) after 3–5 min of deep-frying at 180 °C. Additionally, sensory score results have shown that fried pork skins with 4 mm raw skin thickness had lower flavor, texture, and overall acceptability than those with 2 mm and 3 mm raw skin thickness. Scanning electron micrographs (SEM) have revealed less holes and irregular and crack microstructure in fried pork skins with 4 mm raw skin thickness than in other groups. Different thickness of raw pork skins resulted in different effects in microstructure and influenced water evaporation and oil uptake of fried pork skin. Finally, we have proposed the kinetic equations of water loss and oil uptake of fried pork skins. Fried pork skin from raw skin thicker than 4 mm need frying at temperature higher than 180 °C to improve their puffing ratio and sensory acceptability.

## 1. Introduction

Pork skins are one of the by-products (e.g., skins, hairs, viscera, heads, bones) in the pork processing industry, contributing around 3–8% of the total pork weight [1]. As such, several approaches to turn pork skin into value-added food products have been reported. Since pork skin contains large amounts of collagen and gelatin, it is considered as the most common raw protein material, especially for large-scale extraction [2]. For example, Kaeb Moo is a fried pork skin that is seasoned or dipped in soup and marketed as a snack in Asian countries. The production of this very popular traditional food is estimated at 100 million US$ in 2011 in Northern Thailand [3].

Unfortunately, the oil/fat content of fried pork skins is around 35.71% of the product, making it an unhealthy snack [4]. Despite the health risks related to consuming fried foods, such as diabetes, cardiovascular disease, hypertension, and obesity, the consumption of fried food continues to increase [5], as fried snack lovers and consumers of different ages continue eating their favorite fried foods [6].

Frying is a very old and simple cooking method with very sophisticated cooking mechanisms [7]. During traditional deep-frying, pork skins lose water and absorb oil/fat while undergoing other chemical and physical changes, such as oxidation, protein denaturation, water activity reduction, polymerization, sterilization, and hydrogenation, to form the crispy skin [8]. Frying at high temperatures, usually 150–200 °C, generates aroma, called reaction flavor, which gives fried food its rich taste, color, and texture. Information related to raw pork skin thickness and its influence on fried pork skins is limited.

This study aims to evaluate the influences of raw pork skin thickness on moisture loss and oil uptake using traditional frying approaches to optimize process conditions. We studied raw pork skins with different thickness in terms of quality attributes, such as moisture, oil content, texture, color, thickness, sensory scores, kinetics of thermal, and mass transfer, as well the appearance of fried pork skins, in order to determine different properties of fried pork skins.

## 2. Materials and Methods

### 2.1. Sample Preparation

Frozen skin of pork (*Sus scrofa domestica*) was obtained from Tai An Food Enterprise Co., Ltd. (Kaohsiung City, Taiwan). Frozen pork skin samples were placed in running tap water until they had completely thawed and cooked in boiling water for 1 h. Inner skin fat was scraped to get 0.2 cm, 0.3 cm, and 0.4 cm thickness and then cut the skin into 5 cm × 2 cm strips. Each sample preparation was performed in triplicate with three sub-samples per replication. These strips were dried at 50 °C for 12 h and sealed in polyethene (PE) bags at room temperature until frying within 3 days.

### 2.2. Traditional Frying

Deep-fried the dried pork skin (30 pieces around 60 g each) was deep-fried in refined palm olein oil using an electric fryer (WFT-8L, WISE Co., Ltd., Taipei, Taiwan) at 180 °C [9]. At different time intervals (0.5, 1, 1.5, 2, 3, 4, and 5 min), at least three pork skin samples were taken out and examined individually at each replication.

### 2.3. Moisture and Lipid Content

The moisture content of fried pork skins was calculated following the AOAC method 984.25 by using a convection oven to dry the samples at 105 °C to a constant weight. The wb in this study was determined based on pork skin not being dried to remove water while the dry weight basis (db) is based on pork skin samples dried at 105 °C to a constant weight. The lipid content of the fried pork skin samples was calculated following the diethyl ether extraction method using a Tecator Soxtec System HT1043 (Foss Analytical Co., Ltd., Hillerod, Denmark) at 96 °C for 4 h [10].

### 2.4. Mathematical Modelling

A uniform water distribution within each dried pork skin and constant oil temperature during frying was assumed in the samples. Additionally, we considered water and oil as independent variables. The mathematical modeling of moisture during frying of pork skin was proposed as a first order kinetic model as described by Krokida et al. [11], who explained the mass transfer phenomena in potato strips during frying using the formula below.
(X − X_min_)/(X_0_ − X_min_) = exp (−K_x_t).(1)
t: frying time. X: the moisture content at t (g/g db). X_min_: the moisture content at an infinite t (g/g db). X_0_: the moisture content at t = 0 (g/g db). K_x_: the rate constant of moisture loss (min^−1^).

Similarly, the oil uptake kinetic model during frying of pork skin was proposed as first order, with modifications from the equation proposed by Krokida et al. [11].
Y = (Y_max_ − Y_0_) [1 − exp (−K_y_t)] + Y_0_.(2)
t: frying time (min). Y: the oil content at t (g/g db). Y_max_: the oil content at an infinite t (g/g db). Y_0_: the oil content at t = 0 (g/g db). K_y_: the rate constant of oil absorption (min^−1^).

### 2.5. Puffing Ratio

The puffing ratio of fried pork skins was calculated as the difference between the volumes before and after frying using the sesame seed displacement method described by Chen et al. [12].

### 2.6. Color

The color of fried pork skins was recorded using a spectrocolorimeter (TC-1800 MK II, Tokyo, Japan) following the method described by Le et al. [13].

### 2.7. Breaking Force of Fried Pork Skins

The breaking force of fried pork skins was tested following the method described by Su et al. [14].

### 2.8. Sensory Evaluation of Fried Pork Skins

The fried pork skins with different raw pork skin thickness (labeled with three-digit codes) and commercial products with similar moisture contents were served to untrained sensory panelists who will evaluate the aroma, color, texture, greasy intensity, flavor, and overall scores. Students were instructed to evaluate each attribute using a nine-point hedonic scale, ranging from ‘1 = dislike extremely’ to ‘9 = like extremely.’ Twenty-three male and forty-two female students and faculty members (age: 20–43 years old) of the Department of Food Science participated in the study. Each data point from the sensory analysis represents an average of 65 panelists.

### 2.9. Scanning Electron Microscopy

The fried pork skins were cut using a razor blade after lipid content testing, placed on brass stubs, and sputter-coated with platinum for 30 s (Hitachi E-1010, Tokyo, Japan). The samples were examined under a scanning electron microscope (Hitachi HR-FESEM S-4100, Tokyo, Japan) at 15 kV (50 and 3000× magnification) [15].

### 2.10. Statistical Analysis

The data were analyzed using one-way analysis of variance and tested the general linear model using the Statistical Package for the Social Science (SPSS v.23.0) for Windows (SPSS Inc., Chicago, IL, USA). Ducan’s multiple range test was carried out to detect differences in means at a 5% significance level (*p* < 0.05). The mathematical modeling of oil and moisture content kinetics were established by the residual sum of squares using Statgraphics 18 software (Statgraphics Technologies, Inc., The Plains, VA, USA). Linear correlation was tested by Pearson correlation analysis at *p* < 0.01 significance level.

## 3. Results

### 3.1. Moisture Content and Kinetic Model of Water Loss 

The initial moisture contents of dried pork skins with raw pork skin thickness of 2 mm, 3 mm, and 4 mm were 9.20%, 11.21%, and 13.07% (% wb), respectively. These quickly decreased to 1.75%, 1.68%, and 1.98% during the first 3 min, 4 min, and 5 min of deep-frying, respectively, due to the optimal frying time of fried pork skins (Figure 1). We noted a quick decline in moisture content in thinner pork skin than that in thicker pork skin, which reached around 1.75% in 3 min. The final moisture content of all samples obtained in this study is similar to the results (1.62–1.93%) of local producers in fresh street market and laboratory [16].

The frying oil penetrated into the thicker skin quickly, removing 30% of the initial moisture content within 30 s. Dehydration slowed down and reached around 1.68–1.75% in 3 min and 1.33% in 5 min (Figure 1). The initial oil content of dried pork skins was around 3.17–5.48% (Figure 2). We found that the optimal frying time for fried pork skins is at 3–5 min. The 2 mm thick pork skin needed 3 min of deep-frying with 1.8% wb moisture and 42.72% db oil content, which was almost similar to the 4 mm thick pork skin that needed 5 min of frying with 1.98% wb moisture and 39.49% oil content (Table 1). We also found that the oil content of fried pork skin was between 39.49–42.72% under different raw pork skin thickness (2 mm–4 mm) (Figure 2) at 180 °C.

### 3.2. Physical Properties of Fried Pork Skins

Table 1 lists the ideal frying time for pork skins with different thickness. Water content, oil content, breaking force, and puffing ratio of fried pork skin showed no significant differences. On the other hand, the thickness of raw pork skin (0.2–0.4 cm) did not significantly influence these physical properties. However, deep-fried pork skins from 0.4 cm raw skin thickness had higher final thickness after frying than those form thinner raw pork skin (Table 1).

As shown in Appendix A and Table 2, the data from Krokida’s kinetic model equation and water loss parameters matched the theoretical values. The rate constant of moisture loss (Kx, min^−1^) in the 4 mm thick raw pork skin was lower than that of the 2 mm thick raw pork skin (Table 1), indicating a slower mass transfer of water due to a slower heat transfer in thicker fried pork skin. The moisture content at the infinite frying time should be close to zero. Deep-frying the 2 mm, 3 mm, and 4 mm raw pork skins resulted in 0.02, 0.02, and 0.03 (g/g db) moisture content, respectively, which were ideal for frying since the equilibrium moisture content increases as raw pork skin thickness increase. The final product from the 2 mm and 3 mm raw skin had a lower value of 0.02 (X_min_) than that from the 4 mm raw skin (0.03, Table 2) since the rate constant of moisture loss and mass transfer of water in the thicker raw pork skin was lower. The effective diffusion coefficient of raw pork skin also confirmed this result (Table 2). As predicted by Krokida’s model equation, each model presented a value of R^2^ > 0.9819, indicating that the high coefficient of determination for traditional frying process was close to one (Table 2). In terms of moisture loss, the parameter estimation results for frying pork skins are similar to those mentioned for French fries [11].

### 3.3. Oil Absorption and Kinetic Model of Oil Uptake

Figure 2 shows the oil content of deep-fried pork skins with different raw skin thickness. The moisture content of dried pork skins decreased (from 9.20–13.07% wb to 1.75–1.98% wb) after 5 min of frying, while oil content increased (from 3.17–5.19% db to 49.17–39.49% db). The oil content varied between 3.17% and 49.17% db for fried pork skin with different frying time (0–5 min) (Figure 2).

### 3.4. Changes in the Breaking Force, Puffing Ratio, and Color during Frying

Frying a thinner raw pork skin resulted in a lower breaking force of the product (Figure 3 and Table 1 and Table 3). Figure 3 and Table 3 showed the breaking force, which is related to the product’s crispiness and hardness, and puffing ratio of fried pork skins during different frying time. However, results revealed that the raw pork skin thickness (2–4 mm) did not significantly affect the breaking force of fried pork skins (Figure 3 and Table 1).

Table 3 showed the puffing ratio of fried pork skins with different frying time and raw pork skin thickness. Results revealed that the maximum puffing ratio for the 2 mm thick raw pork skin was reached after 0.5 min of frying. On the other hand, the maximum puffing ratio for the 3 mm and 4 mm thick raw pork skins was reached after 1 min and 2 min of frying, respectively (Table 3).

The puffing ratio of fried tilapia skin increased with frying time [17]. In contrast, the puffing ratio of fried pork skins already increased within the first 2 min of frying but then decreased slightly after optimal cooking time (Table 3).

In this study, fried pork skins showed higher L* and b* values and lower a* values after 0.5 min of frying (Table 4), possibly due to the 200% puffing ratio of fried pork skin and its porous surface structure, increasing the lightness and yellowness of the fried pork skin (Table 3 and Table 4).

### 3.5. Microstructure of Fried Pork Skins

As shown in Figure 4a–d, we observed irregular and non-uniform hole structures in deep-fried pork skins from 2 mm and 3 mm thick raw skins, respectively. We believe that spontaneous evaporation, protein denaturation, dehydration, collagen gelatinization, and absorption of frying oil during frying created these structures. The SEM of deep-fried pork skin from 4 mm thick raw pork skin (Figure 4e,f) showed that the product retained its regular shape, integrity, and compact structure compared with the other sample groups since the drying process forms a compact structure after boiling. This implied that the slower heat and mass transfer during frying caused by the thicker raw pork skin resulted in a uniform structure.

### 3.6. Sensory Score of Fried Pork Skin

Sensory scores (appearance, odor, flavor, texture, greasy intensity, and overall acceptability) of fried pork skins with 0.2 cm and 0.3 cm thick raw pork skin showed no significant difference (Table 5). On the other hand, the flavor, texture, and overall acceptability of fried pork skins from 0.4 cm thick raw pork skin were lower than those of the other two groups (*p* < 0.05), possibly because of its thicker and harder pork skin and because they did not puff very well.

### 3.7. Correlation between Physical Characteristics and Sensory Score of Fried Pork Skin

Correlation analysis was conducted to better understand the relationships among different quality attributes (Appendix A). Frying time (not including t = 0) was negatively correlated with breaking force, moisture content, and water activity, but it was positively correlated with oil content and raw pork skin thickness (*p* < 0.01). If the frying time including t = 0, the correlation with moisture content, oil content, breaking force, and raw skin thickness would increase. It may be due to the fact that the moisture content of dried pork skin has been dramatically decreased, and it would increase the correlation between these parameters. Puffed ratio of fried pork skin did not change with frying time; therefore, the puffed ratio showed no correlation with frying time in fried pork skin. Under similar moisture content ranges, raw pork skin thickness was positively associated with breaking force and thickness after frying (*p* < 0.01). There is no different between the sensory scores (appearance, odor, flavor, texture, greasy intensity, and overall acceptability) of fried pork skin with 0.2 cm and 0.3 cm raw pork skin thickness (Appendix A). Nevertheless, the flavor, texture, and overall acceptability of fried pork rinds with 0.4 cm raw pork skin thickness is lower than the other two tested groups (*p* < 0.05). It may be due to the pork skin is thicker and harder than the other two test groups, and they are not puffed very well, to downgrade these ratings.

The correlation of sensory attributes of fried pork skin is shown at Appendix A. There is high correlation between flavor, texture and overall acceptability of fried pork skin, and it indicates the desirable quality characteristics, such as aroma, flavor, and texture, were associated with consumer perception, it was deserved and needed to be investigated to make a pork skin snack with good taste and aroma but low fat content. There is no significantly correlation greasy intensity with appearance, aroma, flavor, and texture of fried pork skin (Appendix A).

## 4. Discussion

### 4.1. Moisture Content and Kinetic Model of Water Loss

Water evaporation and oil absorption are the result of drying during frying. Moisture in pork skin is lost when water droplets leave the skin surface and evaporate during frying due to high oil temperatures (>100 °C). During the early stages of frying, we observed an intense bubbling over the skin surface. As the frying time approaches 3 min, the process of losing moisture in the 2 mm and 3 mm pork skin changes (Figure 1). Since most of the surface water has rapidly evaporated, the internal skin cell structure forms a layer to slow down the moisture loss and oil absorption. We noticed that the moisture evaporation rate remained approximately constant for 3–5 min while frying the 2 mm and 3 mm pork skin. Because of the structural changes and reductions in the absorption of frying oil into the inner skin, we could ignore the limited rate of convection of oil at this time. As the water in the inner skin tissues evaporates, the superheated water and steam increases the pressure inside the skins. As the inhomogeneity of the pork skin tissue forms blister on the skin surface, the skin becomes crispy because of the continuous evaporation of moisture and denaturation of proteins. Because the 4 mm pork skin is thicker, its moisture content is still higher than that of the 2 mm and 3 mm pork skins after 3 min of frying (Figure 1).

In comparison, deep-fried French fries are thicker, so they are divided into two dimensions: a moist core with high moisture content and a thin dried crust with moisture content less than 5%. Similarly, potato slices and fish skin less than 3 mm thick have a high specific surface area, high dehydration rates, and heat transfer during frying, resulting to rapid changes in texture and high oil uptake [17]. In the case of potato chips, fried fish skin, and fried pork skins, a moving vaporization front surrounds the crust and rapidly reaches the geometric center of the slice or skin, maintaining the moisture loss and oil absorption. This triggers the transport mechanism of water and steam, as well as other mechanisms, such as mass transfer, physicochemical reaction and transformation, and diffusive and convective heat transfer, as the core temperature reaches above boiling point during deep-frying. However, for starchy products, starch undergoes gelatinization first and then dehydration through moisture evaporation, which is influenced by the slice thickness, water content of raw material, oil temperature, and frying time. On the other hand, fried pork skin undergoes protein coagulation, crust and blister formation, and moisture evaporation.

The moisture in the dried pork skin played an important role as it evaporated during the frying process. The steam cooked the denatured pork skin as it escaped through the pores because of internal pressure. When the fried pork skin was taken out of the hot oil, steam production was reduced, and the skin started to cool down. As the internal pressure decreased, the voids created by water evaporation through the pores absorbed the cooking oil into the outer layer.

### 4.2. Physical Properties of Fried Pork Skins

In comparison, 1 mm thick tilapia skin with 68% moisture content deep-fried for 6 min showed significantly lower puffing ratio (292%) and higher oil content (47.6%) [17] than deep-fried pork skins (puffing ratio: 366–425%; oil content: 39.5–42.7%) (Table 1). Since oil content in fried food might be related to their initial moisture content, composition, and structure, we believe that skin porosity of pork and fish during frying plays an important role in the subsequent oil uptake. Additionally, we found no significant difference in breaking force and oil content of the three fried pork skin samples, although thinner raw pork skin seemed to contain more oil and less breaking force and thickness after frying (Table 1). On the other hand, since raw tilapia skin is thinner (1 mm) and has higher moisture content (68%), it requires 6 min of frying, much longer than what is needed for dried pork skin (9.2–13.1% moisture content) in this study (3–5 min). Therefore, we estimated that tilapia skin should be fried for 6 min to reach desired texture (breaking force 14.7 N) and water content (2.4% moisture content) as a snack. In contrast, the puffing ratio of pork skins with 4 mm raw skin thickness is not high enough if fried for less than 5 min, unless these are cooked under higher frying temperatures. Previous studies noted that the puffing ratio (292.4%) of tilapia skin deep-fried for 6 min continued to increase to 318% at the 8-min mark [17]. This ratio was still lower than that of the fried pork skin samples (324–425%) (Table 1).

### 4.3. Oil Absorption and Kinetic Model of Oil Uptake

Figure 2 shows that an increase in thickness of raw pork skin reduced oil uptake after 1 min of frying, probably because the moisture in thicker skin (4 mm) evaporates slower than that in thinner skins (2 mm and 3 mm) during frying at the same temperature. Oil absorption is strongly influenced by the viscosity of the frying oil and the microstructure and surface characteristics of the fried food. As the pork skin is fried for a longer time, more moisture evaporates from the inner skin tissues, and more holes and cracks are created, increasing oil absorption when the food is removed from the hot oil. The hole size, depth, and smoothness of the pork skin surface also influence the oil absorption during and after frying. In order for the steam to continue to flow, the crust has to remain permeable so that moisture can be transported from the inner skin to the crust. As moisture leaves voids in the skin, capillary gaps are created where the oil enters the inner skin, which is why oil uptake peaks during the first 30 s of frying (Figure 1 and Figure 2).

Since the inner skin contains more fat than the outer skin, scraping the inner skin to 0.2 cm thickness in raw pork skin resulted in less fat content (3.17%) compared with those scraped to 0.3–0.4 cm thickness (5.48–5.19%). De Oliver Faria et al. [18] reported that the average pork skin comprised 55.12% moisture, 36.2% protein, and 3.02% fat. Therefore, the oil content of fried pork skins almost completely replaced the moisture loss due to evaporation and the original fat content. In the case of 0.2 cm thickness raw pork skin, since the skin is thinner and contained less oil, its oil content reached maximum values at 49.17% after 5 min of frying (Figure 2).

As mentioned earlier, the mathematical model for oil absorption in pork skin was adapted from the first order oil absorption model for frying potatoes, as described by Krokida et al. [11]. In their experiments, the initial oil content was assumed to be zero. However, in this study, the oil content of dried pork skin was not zero (3.17–5.19% db). Hence, we modified the proposed equation as shown in Equation (2). As shown in Appendix A, the coefficient of determination (R^2^) was greater than 0.98 (Table 2), indicating that the predicted values from the first order oil absorption kinetic model matched the collected data. 

### 4.4. Changes in the Breaking Force, Puffing Ratio, and Color during Frying

Previous research [14] reported that the initial thickness of potato slices significantly affected the hardness of potato chips: the hardness of potato chips positively correlated with the initial thickness. The previous work also found larger porous structures on the surface of thin potato slices. In comparison, the breaking force of fried tilapia skins increased with frying time before reaching 10% moisture content [17], which was in contrast to the 9.20–13.07% wb moisture content of dried pork skin in our study. The breaking force of fried fish skins decreased, while moisture content also decreased from 10% to around 3% as frying continued [17]. However, we did not see a significant increase in the breaking force caused by the denaturation of protein and dehydration as moisture content reached 10%. We did observe a decrease in the breaking force caused by the denaturation of protein among others in the fried pork skin, similar to that in the fish skin. However, the breaking force of fried pork skins did not change significantly after 1 min frying time (Figure 3) since the moisture content was already less than 6%. Therefore, the breaking force of fried pork skin did not change significantly with increased frying time.

However, the optimal puffing ratio for the 3 mm and 4 mm thick raw pork skins reached after 1 min of frying time was not optimal for cooking (Table 1), indicating that the maximum puffing ratio occurred before the optimal frying time of fried pork skins. During deep-frying, denaturation and dehydration formed the pork skin crust, as water evaporated to puff the fried pork skin structure until it reached around 3% moisture content. After reaching the maximum values, the puffing ratio of fried pork skin decreased, possibly because of the increased gelatinization and solubilization of protein due to overheating. Besides having the best puffing ratio in this study, the 2 mm thick raw pork skin also showed better puffing ratio than tilapia skin (1 mm) [17]. Raw tilapia skin only had 292–317% puffing ratio, compared with the deep-fried pork skins’ 324–425% puffing ratio (Table 1). Fang et al. [17] found that vigorous evaporation moved moisture towards the interior of the tilapia skin, while a dehydrated and denatured fish skin formed a crust. As frying continued, the water content became lower, while the evaporation process slowed down. This formed a hollow structure in the fried tilapia skin similar to that in the fried pork skin [17].

When a crust begins to form at the surface of fish skin or pork skin, an excessive pressure builds up, and the fried skin expands and puffs. This explains why the crust of fried pork skins formed earlier than that of fried tilapia skin. Deep-frying raw pork skin or boiled pork skin would not result in puffing. However, deep-frying pre-dried tilapia skin to 9–13% moisture content would result in puffing with a hollow structure. This implies that differences in composition and structure of tilapia skin and pork skin could influence the puffing ratio, oil uptake, and processing method. Frying different parts of pork skin may also influence these factors. Therefore, a better understanding of the relationships between various parameters and the transport processes could control oil uptake and optimize the frying process.

The color of the fried pork skins plays an important role in consumers’ perception of the snack. The color values and color changes of fried pork skins with different raw skin thickness did not show significant difference during frying, indicating a slight occurrence of heat-triggered Maillard and browning reactions.

## 5. Conclusions

Processing pork by-products, such as fried pork skins, is very resource efficient. This study established the oil absorption and water loss kinetic models during deep-frying of pork skins. High water loss and oil absorption rate of pork skin were found within 1 min frying. We found that frying time, breaking force, moisture content, oil content, and skin thickness after frying have high positive correlations (*p* < 0.05). Moreover, raw pork skin thickness also positively correlated with the breaking force and thickness of fried pork skins (*p* < 0.01). We developed kinetic models in frying pork skins with different raw skin thickness to predict the changes in moisture and oil content during frying. The respective coefficients of determination (R^2^) are all above 0.90, implying that the moisture loss and oil absorption predicted by the first order models fit the research data. When comparing the different raw pork skin thickness, we found that deep-fried pork skin had low skin thickness and high puffing ratio, indicating fast moisture loss and oil absorption during deep-frying. Less holes and irregular and crack microstructure in pork skin with thicker raw skin. However, the fried pork skins from 4 mm thick raw skin had lower rating scores in flavor, texture, and overall acceptability, indicating the need for higher frying temperatures to improve the puffing ratio and sensory acceptability for future industry use.

## Figures and Tables

**Figure 1 foods-10-03029-f001:**
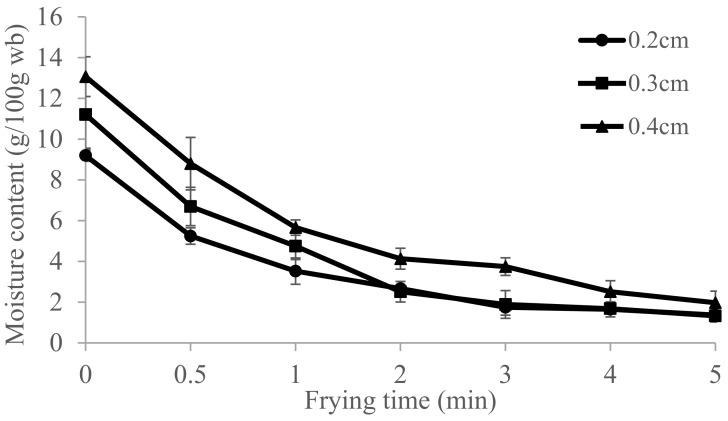
The moisture content (wet basis) of fried pork skins were determined every min for 5 min at 105°C drying using a convection oven. (*n* = 3; *p* < 0.05).

**Figure 2 foods-10-03029-f002:**
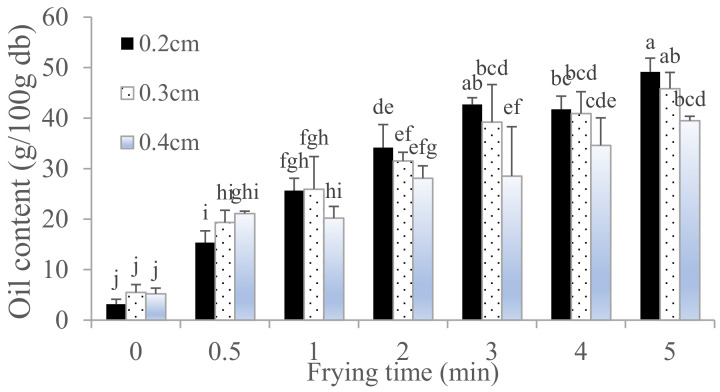
The oil content (dry basis) of fried pork skins were determined every min for 5 min at 105 °C drying using a convection oven. ^a–j^ Indicate significant difference between different samples of fried pork skin. (*n* = 3; *p* < 0.05).

**Figure 3 foods-10-03029-f003:**
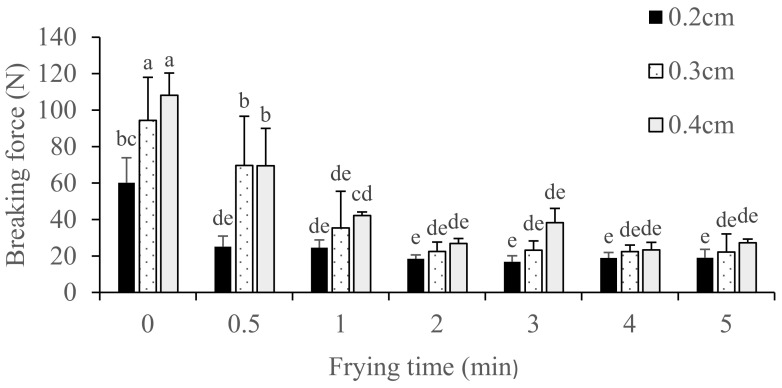
Changes on breaking force of fried pork rind with different thickness and frying time. ^a–e^ Indicate significant difference between different samples of fried pork skin. (*n* = 3; *p* < 0.05).

**Figure 4 foods-10-03029-f004:**
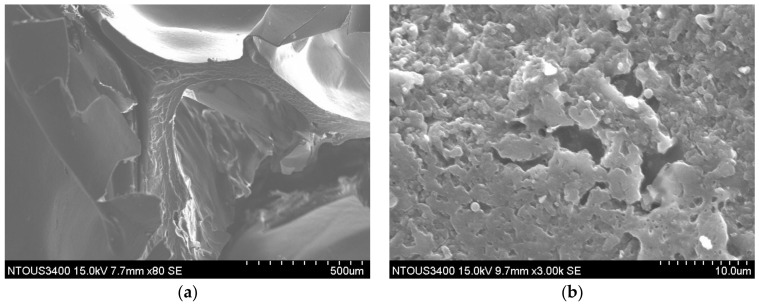
Effect of the raw pork skin thickness in the microstructure of fried pork rinds via scanning electron microscopic analysis (500 μm & 10 μm scale); 0.2 cm dried pork skin (**a**,**b**), 0.3 cm dried pork skin (**c**,**d**), 0.4 cm dried pork skin (**e**,**f**).

**Table 1 foods-10-03029-t001:** The physical properties of fried pork skin with different raw pork skin thickness under the similar water content.

Thickness	Frying Time	Water Content (g/100 g wb)	Oil Content (g/100 g db)	Breaking Force (N)	Puffing Ratio (%)	Thickness after Frying (cm)
0.2 cm	3 min	1.75 ± 0.38	42.72 ± 1.31	16.80 ± 3.33	425.12 ± 77.83	0.44 ± 0.05 ^b^
0.3 cm	4 min	1.68 ± 0.16	40.92 ± 4.32	22.43 ± 3.60	324.18 ± 44.07	0.60 ± 0.07 ^b^
0.4 cm	5 min	1.98 ± 0.06	39.49 ± 0.87	23.72 ± 7.40	365.98 ± 86.50	0.80 ± 0.10 ^a^

Expressed as mean ± standard deviation (*n* = 3). Values followed by different letters within each column are significantly different (*p* < 0.05).

**Table 2 foods-10-03029-t002:** Parameters of moisture loss in Krokida model equation for different frying thickness of pork skin.

Thickness	K_x_	k	D_eff_	X_min_	X_0_	R^2^
0.2 cm	1.46 ± 0.13	0.024 ± 0.002	3.89 × 10^−8^	0.02 ± 0.00	0.10 ± 0.00	0.9916
0.3 cm	1.39 ± 0.13	0.023 ± 0.003	8.39 × 10^−8^	0.02 ± 0.00	0.13 ± 0.02	0.9902
0.4 cm	1.20 ± 0.16	0.020 ± 0.003	1.30 × 10^−7^	0.03 ± 0.00	0.15 ± 0.01	0.9819

Expressed as mean ± standard deviation (*n* = 3). Values followed by different letters within each column are significantly different (*p* < 0.05). K_x_: the rate constant of moisture loss (min^−1^); k: the rate constant of moisture loss (s^−1^); D_eff_: effective diffusion coefficient (m^2^/s); X_min_: the moisture content at infinite frying time (g/g db); X_0_: the moisture content at zero time (g/g db); R^2^: coefficient of determination.

**Table 3 foods-10-03029-t003:** Puffing ratio of fried pork rind with different raw pork skin thickness and frying time.

	0.2 cm	0.3 cm	0.4 cm
30 s	482.76 ± 170.80 ^ab^	296.62 ± 15.94 ^b^	200.6 ± 80.34 ^c^
1 min	535.62 ± 135.24 ^a^	415.28 ± 71.31 ^ab^	350.79 ± 70.80 ^b^
2 min	470.98 ± 109.44 ^ab^	370.45 ± 65.78 ^ab^	440.46 ± 67.44 ^ab^
3 min	448.97 ± 94.10 ^ab^	390.67 ± 88.91 ^ab^	361.42 ± 76.22 ^b^
4 min	467.71 ± 181.57 ^ab^	305.78 ± 87.66 ^b^	433.77 ± 139.63 ^ab^
5 min	381.06 ± 119.89 ^b^	349.54 ± 124.84 ^b^	365.98 ± 86.50 ^b^

Expressed as mean ± standard deviation (*n* = 3). Values followed by different letters within each column are significantly different (*p* < 0.05).

**Table 4 foods-10-03029-t004:** L*a*b* value of fried pork rind with different frying time and raw pork skin thickness.

	Frying Time	L*	a*	b*	ΔE
Dried	-	45.91 ± 3.70 ^g^	−6.06 ± 1.02 ^a^	28.31 ± 1.81 ^g^	-
0.2 cm	30 s	66.13 ± 5.22 ^ef^	−10.41 ± 0.54 ^bcde^	33.87 ± 1.23 ^cdef^	21.49 ± 1.61 ^cd^
	1 min	67.16 ± 3.66 ^def^	−10.29 ± 0.36 ^bcde^	32.26 ± 1.38 ^defg^	22.16 ± 7.42 ^bcd^
	2 min	70.57 ± 1.78 ^abcdef^	−10.54 ± 0.51 ^bcde^	34.36 ± 2.46 ^bcdef^	25.98 ± 5.63 ^abcd^
	3 min	68.68 ± 5.01 ^bcdef^	−10.56 ± 0.61 ^bcde^	34.64 ± 3.92 ^abcdef^	24.28 ± 7.50 ^abcd^
	4 min	74.84 ± 5.53 ^abcd^	−10.05 ± 1.28 ^bcd^	37.22 ± 1.54 ^abcd^	30.56 ± 9.36 ^abc^
	5 min	67.66 ± 2.43 ^cdef^	−9.48 ± 0.76 ^bc^	35.61 ± 4.73 ^abcde^	23.56 ± 6.52 ^abcd^
0.3 cm	30 s	74.52 ± 2.51 ^abcde^	−10.97 ± 1.19 ^bcde^	37.34 ± 1.40 ^abcd^	30.52 ± 3.34 ^abc^
	1 min	76.20 ± 4.67 ^abc^	−11.79 ± 0.91 ^e^	36.13 ± 1.20 ^abcde^	31.86 ± 3.22 ^abc^
	2 min	75.88 ± 2.87 ^abc^	−11.09 ± 0.73 ^cde^	35.19 ± 4.13 ^abcdef^	31.25 ± 4.85 ^abc^
	3 min	77.43 ± 2.71 ^ab^	−11.53 ± -0.86 ^de^	39.54 ± 1.52 ^a^	33.99 ± 2.11 ^a^
	4 min	77.53 ± 8.67 ^a^	−11.65 ± 0.77 ^de^	39.12 ± 2.71 ^ab^	33.97 ± 6.21 ^a^
	5 min	77 ± 4.96 ^ab^	−11.26 ± 1.33 ^de^	38.03 ± 1.20 ^abc^	33.01 ± 4.23 ^ab^
0.4 cm	30 s	73.16 ± 2.25 ^abcde^	−10.28 ± 0.12 ^bcde^	36.49 ± 1.25 ^abcd^	28.79 ± 4.01 ^abcd^
	1 min	66.64 ± 5.91 ^def^	−10.21 ± 0.76 ^bcde^	31.33 ± 3.49 ^efd^	21.46 ± 4.53 ^cd^
	2 min	64.07 ± 6.86 ^f^	−9.42 ± 0.97 ^b^	30.41 ± 3.73 ^fg^	18.79 ± 7.19 ^d^
	3 min	73.91 ± 2.47 ^abcde^	−11.14 ± 0.28 ^de^	33.84 ± 3.20 ^cdef^	29.12 ± 6.69 ^abcd^
	4 min	75.29 ± 3.49 ^abcd^	−10.59 ± 0.91 ^bcde^	37.81 ± 2.56 ^abc^	31.28 ± 7.42 ^abc^
	5 min	73.97 ± 3.46 ^abcde^	−11.24 ± 0.81 ^de^	34.02 ± 2.21 ^bcdef^	29.13 ± 6.32 ^abcd^

Expressed as mean ± standard deviation (*n* = 3). Values followed by different letters within each column are significantly different (*p* < 0.05).

**Table 5 foods-10-03029-t005:** The scores of sensory evaluation with fried pork skin under different thickness.

	Appearance	Odor	Flavor	Texture	Greasy Intensity	Overall Acceptability
0.2 cm	5.63 ± 1.74	5.92 ± 1.76	5.86 ± 1.83 ^a^	5.63 ± 1.70 ^a^	4.60 ± 2.35	5.78 ± 1.68 ^a^
0.3 cm	5.97 ± 1.53	5.86 ± 1.53	6.15 ± 1.72 ^a^	5.92 ± 2.25 ^a^	4.46 ± 2.16	5.98 ± 1.70 ^a^
0.4 cm	6.05 ± 1.58	5.63 ± 1.40	4.77 ± 1.80 ^b^	3.69 ± 2.69 ^b^	4.25 ± 2.04	4.06 ± 1.71 ^b^

Expressed as mean ± standard deviation (*n* = 65). Values followed by different letters within each column are significantly different (*p* < 0.05). Using 9-point hedonic test for Appearance, Flavor, Texture and Overall acceptability: 1 = dislike extremely, 5 = neither like nor dislike, and 9 = like extremely. Using 9-point hedonic test for greasy intensity: 1 = no greasy, and 9 = strong greasy.

## Data Availability

The data presented in this work are available on request from the corresponding author.

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
