# Peer review of "Kinetics of Oil Absorption and Moisture Loss during Deep-Frying of Pork Skin with Different Thickness"

_foods, 2021, doi:10.3390/foods10123029_

Round 1
Reviewer 1 Report
Authors have studied, the influences of raw pork skin thickness on moisture loss and oil uptake using traditional frying approaches to optimize process conditions. Quality attributes like moisture, oil content, texture, colour, thickness, sensory scores, the kinetics of thermal and mass
transfer, as well the appearance of fried pork skins in order to determine different properties of fried pork skins. The structure and orientation of the manuscript is good,
Figures were good, adequate tables.
Elaboration was ok, maybe increased comparison with other references
Conclusions can be elaborated.
English can be improved.
Author Response
Responses to Comments and Suggestions for Authors
Foods
Title: Kinetics of oil absorption and moisture loss during deep-frying of pork skin with different thickness (Foods-1408680)
Dear Reviewer #1
The authors are extremely grateful to anonymous reviewer involved for providing his/her excellent comments and valuable advice in this paper. We have revised the paper based on the reviewer’s comments. We have pleasure in requesting the reviewer to review this paper. Thank you. Your prompt attention to this paper is very much appreciated.
Comments and Suggestions for Authors
Point 1: Authors have studied, the influences of raw pork skin thickness on moisture loss and oil uptake using traditional frying approaches to optimize process conditions. Quality attributes like moisture, oil content, texture, color, thickness, sensory scores, the kinetics of thermal and mass transfer, as well the appearance of fried pork skins in order to determine different properties of fried pork skins. The structure and orientation of the manuscript is good, Figures were good, adequate tables. Elaboration was ok, maybe increased comparison with other references. Conclusions can be elaborated. English can be improved.
Response 1:
We have revised title and rewritten and rechecked the manuscript in our article carefully as red marked texts in the revised manuscript. We add the main thoughts driven from the obtained results from our study in conclusion. Although, we have asked Enago, an editing brand of Crimson Interactive Inc., for English language, grammar and spelling as attached certificate of editing. We asked Dr. Lin to recheck the revised manuscript again. If it is not good enough, Enago promised to help us and editing the revised manuscript again soon. Thanks for the suggestions and we very much appreciate your consideration on this matter. (Please see the revised manuscript).
Yours truly,
Wen-Chieh Sung, Ph.D.
Professor
Department of Food Science
National Taiwan Ocean University

Reviewer 2 Report
- English need to be improved
- Abstract. Lines12-13 "with different raw thickness". It needs to be reformulated
- Abstract. Lines 18-20 what does it mean? The sentence is unclear
- Abstract is too general. There is a lack of main conclusion made based on the performed study
- Introduction. Line 35 and 37. These sentences should be combined
- Introduction lines 46-48. The study did not verify the impact on the fat reduction in final product so the reference is inaccurate.
- Material and Methods. Section 2.2. Traditional frying. There is lack of information about study scheme. How many pieces of pork were in each group (0.2; 0.3 and 0.4). The whole scheme must be described. There is a lack of information to repeat the study.
- Statistics. How many pork skin slices were studied for each skin thickness and for each treatment time (n=?). How many repetition for each method used in the study was made?
- The description of the methods performed in scientific articles is usually impersonal. Please reformulate.
- 10. Results. Fig.1 and Fig.2 shortcuts needs to be expand in the footer. Figures must be self-explenatory.
- Table 1 heading. The expression overall expresions is too general
- Table 1, Table 5. You should not enter letters to the value without statistical differences
- Lines 182-184. The last part of the sentance suppose to be removed (compared with frying time abnd moisture content.
- Lines 187-188. What does mean the optimal puffing ratio. Based on what parameters it was set/chosen?
- Line 202. The authors intenion is unclear. The sentence need to be improved.
- Lines 278-289 This part suppose to be in section 4.3
- Section 4.5. There is no discussion here. What is the point of this paragraph?
- Concluisons. Lack of the main thoughts driven from the obtained results
Author Response
Responses to Comments and Suggestions for Authors
Foods
Title: Kinetics of oil absorption and moisture loss during deep-frying of pork skin with different thickness (Foods-1408680)
Dear Reviewer #2
The authors are extremely grateful to anonymous reviewer’s involved for providing his/her excellent comments and valuable advice in this paper. We have revised the paper based on the reviewer’s comments. We have pleasure in requesting the reviewer’s to review this paper. Thank you. Your prompt attention to this paper is very much appreciated.
Comments and Suggestions for Authors
Point 1: English need to be improved.
Response 1: We have revised title and rewritten and rechecked the manuscript in our article carefully as red marked texts in the revised manuscript. Although, we have asked Enago, an editing brand of Crimson Interactive Inc., for English language, grammar and spelling as attached certificate of editing. We asked Dr. Lin to recheck the revised manuscript again. If it is not good enough, Enago promised to help and edit the revised manuscript again soon. Thanks for the suggestions and we very much appreciate your consideration on this matter. (Please see the revised manuscript).
Point 2: Abstract. Lines12-13 "with different raw thickness". It needs to be reformulated
Response 2: The phrase "with different raw thickness" in lines 12-13 was revised to "with different raw skin thickness". Thanks for the suggestion.
Point 3: Abstract. Lines 18-20 what does it mean? The sentence is unclear
Response 3: The sentence in lines 18-20 is unclear and revised to “Different thickness of raw pork skins resulted in different effects in microstructure and influenced water evaporation and oil uptake of fried pork skin”. Thanks for indicating the problem.
Point 4: Abstract is too general. There is a lack of main conclusion made based on the performed study
Response 4: We added the sentence “Fried pork skin from raw skin thicker than 4 mm needs frying at temperature higher than 180°C to improve their puffing ratio and sensory acceptability” on abstract for future research and added several sentences on conclusion section based on the performed study. Thanks for pointing out the problems at abstract and conclusion.
Point 5: Introduction. Line 35 and 37. These sentences should be combined
Response 5: We combined these two sentences and reorganized the reference order as the sentence “Despite the health risks related to consuming fried foods such as diabetes, cardiovascular disease, hypertension and obesity, the consumption of fried food continues to increase [5], fried snack lovers and consumers of different ages continue eating their favorite fried foods [6]” at the second paragraph of the introduction. (Please see the revised introduction at page 1).
Point 6: Introduction lines 46-48. The study did not verify the impact on the fat reduction in final product so the reference is inaccurate.
Response 6: Sentences in lines 46-48 was deleted and the last sentence of the paragraph was moved to previous paragraph. Thanks for reminding the problem.
Point 7: Material and Methods. Section 2.2. Traditional frying. There is lack of information about study scheme. How many pieces of pork were in each group (0.2; 0.3 and 0.4). The whole scheme must be described. There is a lack of information to repeat the study.
Response 7: We add a few information about study scheme at section 2.1 to help the reader could prepare the pork skin sample and repeat this study.
Point 8: Statistics. How many pork skin slices were studied for each skin thickness and for each treatment time (n=?). How many repetition for each method used in the study was made?
Response 8: Each sample preparation and frying time interval was performed in triplicate. At least 3 pork skin slices were measured at thickness and for each frying time (n=3).
Point 9: The description of the methods performed in scientific articles is usually impersonal. Please reformulate.
Response 9: The description of the Materials and Methods was reformulated the personal sentences. Thanks for point the problems and please see the revised Materials and Methods section at page 2 to page 3.
Point 10: Results. Fig. 1 and Fig. 2 shortcuts needs to be expand in the footer. Figures must be self-explanatory.
Response 10: The explanatory footnote was added to Figures 1 & 2. Please see page 4 of the revised manuscript.
Point 11: Table 1 heading. The expression overall expressions is too general
Response 11: Table 1 heading was revised to“The physical properties of fried pork skin with different raw pork skin thickness under the similar water content”. Thanks for point out the problem.
Point 12: Table 1, Table 5. You should not enter letters to the value without statistical differences
Response 12: The upper case letters of data values without statistical differences of Tables 1 and 5 were removed. Thanks for the suggestion.
Point 13: Lines 182-184. The last part of the sentence supposes to be removed (compared with frying time and moisture content).
Response 13: The last part of the sentence at lines 182-184 was deleted.
Point 14: Lines 187-188. What does mean the optimal puffing ratio. Based on what parameters it was set/chosen?
Response 14: The phrase “optimal puffing ratio” was revised to “maximum puffing ratio”. Thanks for point out the ambiguity.
Point 15: Line 202. The authors intension is unclear. The sentence need to be improved.
Response 15: The first sentence at section 3.5 was deleted.
Point 16: Lines 278-289 This part supposes to be in section 4.3
Response 16: The paragraph in lines 278-289 was moved to first paragraph of 4.3. Thanks for the suggestion.
Point 17: Section 4.5. There is no discussion here. What is the point of this paragraph?
Response 17: The texts of section 4.5 were moved to the last paragraph of section 3.7 due to no discussion. Thanks for point out the problem.
Point 18: Conclusions. Lack of the main thoughts driven from the obtained results
Response 18: Two main thought sentences driven from the obtained results were added to conclusion. Please see the conclusion of the revised manuscript at page 12.
Yours truly,
Wen-Chieh Sung, Ph.D.
Professor
Department of Food Science
National Taiwan Ocean University

Round 2
Reviewer 2 Report
The proper changes has been made. However, the data obtained during the experiment and presented in the submitted manuscript is of low scientific interest.